biochemistry/cellular biology/theoretical biology

glutamate/glutamine ratio, epidermis, hypoxic cell environment, epidermal differentiation complex, liver

**Author for correspondence:**
Francesca Silvagno
e-mail: francesca.silvagno@unito.it

†These authors contributed equally to this work.

# The analysis of glutamate and glutamine frequencies in human proteins as marker of tissue oxygenation

Annamaria Vernone[1,†], Chiara Ricca[1,†], Daniela Merlo[2], Gianpiero Pescarmona[1] and Francesca Silvagno[1]

[1]Department of Oncology, University of Torino, Via Santena 5 bis, 10126 Torino, Italy
[2]Department of Neuroscience, Istituto Superiore di Sanità, Viale Regina Elena 299, 00161 Rome, Italy

FS, 0000-0002-8800-9135

In this study, we investigated whether the relative abundance of glutamate and glutamine in human proteins reflects the availability of these amino acids (AAs) dictated by the cellular context. In particular, because hypoxia increases the conversion of glutamate to glutamine, we hypothesized that the ratio glutamate/glutamine could be related to tissue oxygenation. By histological, biochemical and genetic evaluation, we identified proteins expressed selectively by distinct cellular populations that are exposed in the same tissue to high or low oxygenation, or proteins codified by different chromosomal loci. Our biochemical assessment was implemented by software tools that calculated the absolute and the relative frequencies of all AAs contained in the proteins. Moreover, an agglomerative hierarchical cluster analysis was performed. In the skin model that has a strictly local metabolism, we demonstrated that the ratio glutamate/glutamine of the selected proteins was directly proportional to oxygenation. Accordingly, the proteins codified by the epidermal differentiation complex in the region 1q21.3 and by the lipase clustering region 10q23.31 showed a significantly lower ratio glutamate/glutamine compared with the nearby regions of the same chromosome. Overall, our results demonstrate that the estimation of glutamate/glutamine ratio can give information on tissue oxygenation and could be exploited as marker of hypoxia, a condition common to several pathologies.

## 1. Introduction

The local availability of amino acids (AAs) may be a limiting factor in protein biosynthesis. The regulation of the expression

of a protein can occur at different levels, from the epigenetic control of transcription to the stabilization of the messenger, and finally, it relies on coupling transcription with translation. The last step is critical, since the half-life of mRNA is limited and therefore transcription and translation must be coordinated. During AA starvation, the translational regulation has a major influence on gene expression, as demonstrated in simple models [1]. When the levels of free AAs are low, the percentage of some complexes tRNA/AA can become the main limiting factor in protein biosynthesis, because the short half-life of the mRNA does not allow the system to adjust the AA concentration rapidly enough. In this context, the relative abundance of some AAs can be considered a regulatory factor and it can be said that the AA availability can influence gene expression in addition to other mechanisms such as the epigenetic control. To date, the effect of local AA availability in the regulation of protein expression has been scarcely investigated in humans.

In our previous study [2], we found several evidences that the composition of proteins gives information on AA availability and local control of protein expression. The study compared human proteins on the basis of their AA content by a hierarchical, agglomerative clustering method. Keratins and collagens, proteins with a repetitive structure, clustered well, confirming the mathematical approach. The software tool used in the study identified similarities between proteins on the same chromosome, like a group of aquaporins, and divergences between those on different chromosomes, which were more distant in the cluster. Proteins with similar tissue expression, even though with different function, were very close in the cluster, and this was suggestive that the local availability of AAs was a major driver of protein expression, conditioned by tissue context.

The present work deepens this approach based on the evaluation of AA frequencies and focuses on the analysis of glutamate (Glu) and glutamine (Gln), which are strictly linked by a biosynthetic interconversion; starting from our previous observations, we were interested in their relative abundance in protein expressed in different conditions or distinct phases of a cellular life cycle.

The biosynthesis of glutamine from glutamate amidation is catalysed by the enzyme glutamine synthetase (GS) in the following reaction

$$Glu + NH_3 + ATP \leftrightarrow Gln + ADP + P_i.$$

In mammalians, the reaction produces glutamine because the forward rate relative to the reverse rate is about 10 to 1 [3]. The enzyme has been extensively studied in the brain, where the site of glutamine synthesis is in glial cells [4], and it has been investigated also in the liver and muscle [5–7]. GS is also present in the epidermis, strongly expressed in the outer granular layer [8]. The exposure to hypoxia increases both GS enzymatic activity, as demonstrated *in vivo* in the brain [9], liver and muscle [10], and GS mRNA and protein levels, as reported by a study *in vitro* on PC12 cells [11].

The reverse reaction of GS is catalysed by the enzyme glutaminase, which in the brain is predominantly a neuronal enzyme, in the following reaction

$$Gln + H_2O \rightarrow Glu + NH_4^+.$$

The phosphate-activated glutaminase is inhibited by chronic hypoxia, at the levels of mRNA, protein and enzyme activity [11].

Because the hypoxic condition induces the biosynthesis and reduces the catabolism of glutamine, the endogenous levels of glutamine are higher when oxygen supply is scarce.

It is of note that the two enzymes are often spatially separated in tissues; for example, in the brain, they are differentially expressed in neurons and glia [12–14], and in the liver, their expression is opposite between the periportal and perivenous zone [15,16]. In the epidermis, GS is highly expressed in the granular layer, whereas glutaminase has not been characterized in terms of localization. The differential expression not only avoids a futile cycle but also it has generally a homeostatic function in tissue metabolism, with the aim of recycling the glutamate or replenishing the glutamine levels where and when they are low.

The biosynthesis of proteins is conditioned by the local availability of single AAs. The concentration becomes particularly critical for those AAs in equilibrium between synthesis and demolition, such as the case of glutamine; this AA can become rate-limiting in protein biosynthesis and the proteins abundant in glutamine are preferentially synthesized when the local conditions favour glutamine synthesis over catabolism. Based on these considerations, we put forward the hypothesis that the hypoxic condition could promote the translation of proteins richer in glutamine, whereas a more oxygenated environment would advantage the availability of glutamate and therefore the biosynthesis of proteins enriched in glutamate. Every time a biological signal is originated from the concentration changes of

two organic species interconverted by a chemical reaction, the cellular system senses the ratio between the two molecules rather than the single concentrations; for example, the intracellular energy levels are checked by evaluating the cytosolic ATP/AMP or the mitochondrial ATP/ADP ratio.

In our study, we calculated the percentage of Glu and Gln of selected proteins and we tested the hypothesis that the ratio Glu/Gln is directly proportional to oxygenation. We expected that the proteins with a reduced ratio are synthesized in hypoxic condition. We carried out our analysis on two models of tissue that show a gradient of oxygenation: the epidermis and the liver. These tissues are both organized in distinct zones composed by cells exposed to a different environment and able to perform specialized tasks; it is therefore possible to distinguish a population of well-oxygenated cells and a population of poorly oxygenated cells and choose protein markers of the different populations to verify our working hypothesis. However, the two models are different in AA availability from external source and AA utilization other than endogenous metabolism; thus, in our analysis, we took into consideration how the abundance of Glu and Gln is affected by uptake, biosynthesis and loss.

# 2. Experimental methods

## 2.1. Identification of protein markers

Epidermal proteins analysed in this work have been selected from tissue microarray data available in the Human Protein Atlas database (http://www.proteinatlas.org) or from the literature. More precisely, the following proteins were described in some studies as specifically expressed in a single epidermal layer: LFNG, ITA6, KLK14, KLK5, KLK7, PRSS8, PAR2, RFNG [17–21], CADH3 [22], TGM1 [23], K2C5, K1C14, K1C10, H6VRG3, TGM2, INVO, LORI, SPRR4, CRNN, FILA [24], E9PBK1 [25], VDR, CP27B [26]. Liver proteins expressed preferentially in periportal or perivenous epatocytes were selected based on previous works [15,16,27–29]. Proteins coded by EDC, upstream and downstream genes were found by the analysis of chromosome 1q21.3 sequence on the NCBI Map Viewer (https://www.ncbi. nlm.nih.gov/genome/gdv/). The AA sequence of all proteins was downloaded from UniProtKB/ Swiss-Prot. In tables, each protein was identified by the ID (Identification) provided by FLAT files.

## 2.2. Data analysis

The protein sequences in FLAT file format were downloaded from UniProtKB/Swiss-Prot protein database (http://www.uniprot.org/uniprot/). A software tool [2] was applied to the data contained in the FLAT file of the protein. The SQ section of the FLAT file was used to compute the absolute and the relative frequencies of all AAs contained in the protein. The data obtained are presented in the tables of the article and were used for the box plot construction and for cluster analysis.

## 2.3. Cluster analysis

The frequencies tables were used to organize the data in matrix format, with the IDs of the proteins in the rows and the AA relative frequencies in the columns in order to be read using R software (https://www. r-project.org/). An agglomerative hierarchical cluster analysis was performed by using the R package hclust (https://stat.ethz.ch/R-manual/R-devel/library/stats/html/hclust.html). The Ward's method and the Euclidean metric were chosen to calculate the distances between the elements. The results were converted in Newick format using the 'ctc' library from Bioconductor (https://www. bioconductor.org/packages/release/bioc/html/ctc.html) by using R. This format may be imported by the graphical editor Dendroscope (http://dendroscope.org/) in order to visualize the distances between clusters. The clusters' representation was given by means of a horizontal cladogram.

## 2.4. Statistical analysis

All relative frequencies of glutamate, glutamine and their ratio were analysed by GraphPad PRISM (GraphPad Software, San Diego, CA, USA), which was used to perform statistical significance analyses. Statistical analysis was carried out using a Kruskal–Wallis test with the Dunn *post hoc* correction to compare skin's data (three groups). A Mann–Whitney *U*-test was used to analyse liver's data and LIP protein cluster. A *p*-value of less than 0.05 was considered statistically significant.

# 3. Results and discussion

## 3.1. The epidermis model

The epidermis is composed by keratinocytes arranged in layers representative of each step of their programmed life cycle. Proliferating keratinocytes are present on the bottom of the epidermis, called the stratum basale. As keratinocytes leave the basal layer, they begin to differentiate and form the stratum spinosum and then the stratum granulosum, ending in the anucleated stratum corneum composed of dead cornified cells, which has the major role of permeability barrier. Across the layers, the epidermis develops several gradients fundamental for its function. The most studied is the calcium gradient, which ranges from the lowest concentrations in the stratum basale to the highest concentrations in the stratum granulosum where proteins and lipids critical for barrier function are produced. Also, the oxygen supply of the tissue changes drifting away from dermal blood vessels; the most proximal basal layer is more oxygenated than the outer granular layer, the latter being insulated from external atmosphere by the protective epidermal permeability barrier. The epidermal keratinocytes are fed with AAs supplied by dermal circulation, and then they support their specialized functions during migration and differentiation by AA metabolism and biosynthesis. Therefore, we can consider the AA composition of proteins as the indicator of the local availability of the single AA due to the context, for example, due to oxygen levels in that particular epidermal layer. In this model, we searched the proteins expressed selectively by distinct populations of keratinocytes that are exposed to a gradient of oxygen that varies from 21 mm Hg in the basal layer to the tension of 14 mm Hg in the spinous layer and 7 mm Hg found in the granular layer [30,31]. The research of published articles and data banks led to a list of 35 proteins shown in table 1. Then we evaluated their content of glutamate and glutamine as percentage of the whole protein (relative frequency) and as a ratio of percentages (Glu/Gln).

We chose to evaluate the percentage of the total and not the absolute content of Glu and Gln in each protein in order to ignore the differences in protein size. In fact, when proteins have an equal percentage of the AA, the larger the size of the protein, the higher the amount of the AA, but because we do not know the relative expression levels of each protein, we cannot judge how the absolute AA content of a single protein can impact on the intracellular pool of that AA. For example, a protein could have a high number of Gln but a low expression, and another protein could have fewer Gln, but it could be abundantly expressed. Therefore, in our evaluation, the relative frequency describes a general trend of utilization of a certain AA, independently from protein size and expression. Moreover, our results highlighted the importance of the Glu/Gln ratio, in which the relative or absolute content of AA makes no difference.

The relative frequencies of the two AAs were used to investigate the similarity between proteins (cluster analysis) and the relationship between AA content and oxygenation (indicators of oxygenation).

## 3.2. Cluster analysis of epidermal proteins

We investigated the similarities between the selected human proteins by cluster analysis. The cladogram relative to the ratio between glutamate and glutamine highlighted two clear sets for granular and basal proteins, while spinous proteins were distributed over the two different groups. These results are shown in figure 1. Based on the cluster analysis, the Venn diagram shows that granular and basal proteins are grouped in distinct sets, whereas spinous proteins are shared by both sets. By this analysis, we concluded that the two different layers are characterized by a specific environmental context (for example, the oxygenation) able to select the biosynthesis of proteins similar in Glu/Gln ratio. It is reasonable to imagine that because the intermediate spinous layer has a medium degree of differentiation and oxygenation, its proteins share common features with those of both upper and lower layers.

## 3.3. The analysis of the content of glutamate and glutamine and their ratio as indicator of oxygenation of epidermal layers

Because the clustering analysis revealed that the proteins of each layer are similar in Glu/Gln content, we decided to quantify and compare the protein composition of the three layers. The protein markers of the three epidermal layers were evaluated in terms of their content of Glu, Gln and the ratio Glu/Gln (table 1). The median values of arrays were plotted on graph and are shown in figure 2. The layers at the two extremes of the oxygen gradient were significantly different in their content of Glu and Gln: in the hypoxic granular layer, Glu decreased and Gln increased, and most remarkably, the ratio Glu/Gln was

**Table 1.** Summary of selected epidermal proteins divided into different layers. Protein identity (ID) is indicated as found in UniProtKB/Swiss-Prot and the chromosomal location of the relative gene (Chr) is shown.

| layer | protein | ID UniProt | Chr | Glu/Gln (%) | Glu (%) | Gln (%) |
|-------|---------|------------|-----|-------------|---------|---------|
| basal | keratin type II cytoskeletal 5 | K2C5_HUMAN | 12q13.13 | 1.519 | 0.069 | 0.046 |
| basal | vitamin D3 receptor | VDR_HUMAN | 12q13.11 | 1.75 | 0.066 | 0.037 |
| basal | 25-hydroxyvitamin D-1 $\alpha$ hydroxylase, mitochondrial | CP27B_HUMAN | 12q14.1 | 1.75 | 0.069 | 0.039 |
| basal | keratin type I cytoskeletal 14 | K1C14_HUMAN | 17q21.2 | 1.8 | 0.095 | 0.053 |
| basal | transglutaminase 2 | TGM2_HUMAN | 20q11.23 | 2.32 | 0.084 | 0.036 |
| basal | cadherin-3 | CADH3_HUMAN | 16q22.1 | 1.906 | 0.074 | 0.039 |
| basal | $\beta$-1,3-*N*-acetylglucosaminyl transferase lunatic fringe | LFNG_HUMAN | 7p22.3 | 3.6 | 0.047 | 0.013 |
| basal | integrin $\alpha$-6 | ITA6_HUMAN | 2q31.1 | 1.468 | 0.061 | 0.042 |
| basal | collagen $\alpha$-1(XVII) chain | COHA1_HUMAN | 10q25.1 | 1.312 | 0.042 | 0.032 |
| basal | keratin type I cytoskeletal 15 | K1C15_HUMAN | 17q21.2 | 2.042 | 0.107 | 0.053 |
| basal | protein FAM83G | FA83G_HUMAN | 17p11.2 | 1.311 | 0.072 | 0.055 |
| basal | desmocollin-2 | DSC2_HUMAN | 18q12.1 | 1.676 | 0.069 | 0.041 |
| basal | fibroblast growth factor receptor 1 | FGFR1_HUMAN | 8p11.23 | 2.292 | 0.067 | 0.029 |
| basal | B-cell lymphoma/leukaemia 11B transcription factor | BC11B_HUMAN | 14q32.2 | 2.129 | 0.074 | 0.035 |
| spinous | involucrin | INVO_HUMAN | 1q21.3 | 0.773 | 0.198 | 0.256 |
| spinous | keratin 1 | H6VRG3_HUMAN | 12q13.13 | 1.056 | 0.059 | 0.056 |
| spinous | transglutaminase 1 | TGM1_HUMAN | 14q12 | 1.467 | 0.054 | 0.037 |
| spinous | keratin 10 | K1C10_HUMAN | 17q21.2 | 1.692 | 0.075 | 0.045 |
| spinous | dermokine | DMKN_HUMAN | 19q13.12 | 0.87 | 0.042 | 0.048 |
| spinous | galectin-7 | LEG7_HUMAN | 19q13.2 | 1.5 | 0.066 | 0.044 |
| granular | loricrin | LORI_HUMAN | 1q21.3 | 0 | 0 | 0.045 |
| granular | small proline-rich protein 4 | SPRR4_HUMAN | 1q21.3 | 0.043 | 0.013 | 0.291 |
| granular | ATP-binding cassette subfamily A member 12 | E9PBK1_HUMAN | 2q35 | 0.4 | 0.022 | 0.056 |
| granular | cornulin | CRNN_HUMAN | 1q21.3 | 0.629 | 0.089 | 0.141 |
| granular | filaggrin | FILA_HUMAN | 1q21.3 | 0.698 | 0.063 | 0.09 |
| granular | kallikrein-14 | KLK14_HUMAN | 19q13.41 | 0.4 | 0.03 | 0.075 |
| granular | kallikrein-5 | KLK5_HUMAN | 19q13.41 | 0.312 | 0.017 | 0.055 |
| granular | kallikrein-7 | KLK7_HUMAN | 19q13.41 | 0.636 | 0.028 | 0.043 |
| granular | prostasin | PRSS8_HUMAN | 16p11.2 | 0.95 | 0.055 | 0.058 |
| granular | protease-activated receptor 2 | PAR2_HUMAN | 5q13.3 | 0.75 | 0.015 | 0.02 |
| granular | $\beta$-1,3-*N*-acetylglucosaminyl transferase radical fringe | RFNG_HUMAN | 17q25.3 | 0.769 | 0.03 | 0.039 |
| granular | corneodesmosin | CDSN_HUMAN | 6p21.33 | 0.217 | 0.009 | 0.043 |
| granular | Ly6/PLAUR domain-containing protein 5 | LYPD5_HUMAN | 19q13.31 | 0.462 | 0.024 | 0.052 |
| granular | protein POF1B | POF1B_HUMAN | x | 0.912 | 0.088 | 0.097 |
| granular | homeobox protein OTX1 | OTX1_HUMAN | 2p15 | 0.529 | 0.025 | 0.048 |

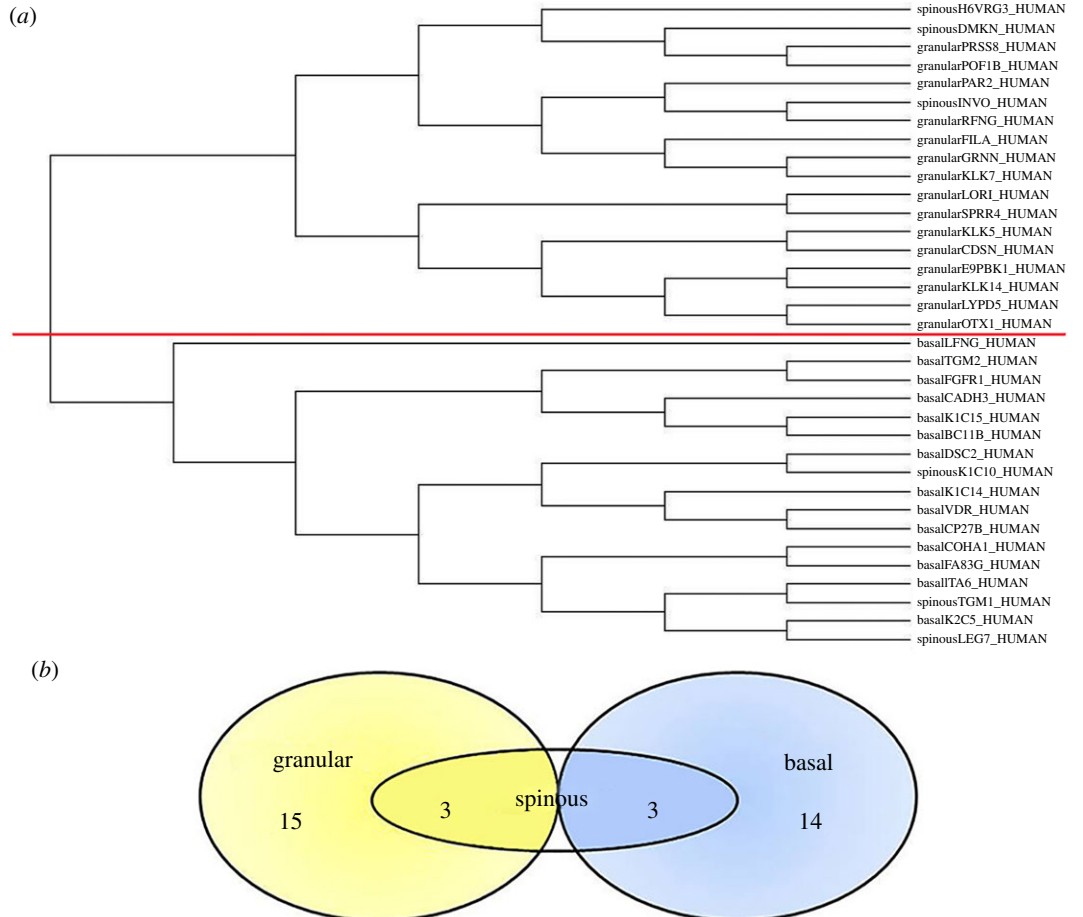

**Figure 1.** Cluster tree of epidermal proteins and relative Venn diagram. (*a*) The tree was created using the ratio Glu/Gln of proteins shown in table 1. Each protein is indicated by the name of the layer (granular, spinous or basal) followed by the ID of the protein. The red line divides the cladogram in two groups, the granular proteins in the upper subtrees and the basal proteins in the lower subtrees, whereas spinous proteins are spread in the whole cladogram. (*b*) Based on the cluster analysis, the Venn diagram shows the distinct sets and the number of proteins that are elements of each group.

significantly lower when granular was compared with the basal layer. Reasonably, the values of the middle spinous layer were intermediate. Based on these data, we concluded that the hypoxic granular layer was able to increase the intracellular levels of glutamine by converting glutamate into glutamine and could afford the synthesis of proteins enriched in this latter AA but sparing the glutamate. As a result, the differences of the ratio Glu/Gln were found to be very significant between basal and granular layers. These observations supported our working hypothesis that the ratio Glu/Gln can be used as a marker of tissue oxygen status.

## 3.4. The investigation of the loci codifying for proteins selectively expressed in granular keratinocytes

Some of the proteins selected as markers of the granular layer are coded by a locus called EDC (epidermal differentiation complex) located on chromosome 1 region q21.3. Actually, the terminal epidermal differentiation programme drives the synthesis of proteins that are codified by EDC [32] and are involved in skin barrier formation: the precursor proteins of the cornified envelope such as involucrin, loricrin and the small proline-rich region (SPRR) proteins, the 'fused gene' proteins such as filaggrin, trichohyalin, hornerin, repetin and cornulin, and the group of calcium-binding proteins (S100) at the edges of the EDC [24,33]; because the EDC proteins are synthesized exclusively by the differentiated hypoxic epidermal granular layer, we considered them as proteins responsive to the same cellular context and therefore to the same AA availability, and we compared them with other two groups of proteins coded by nearby regions of the same chromosome, upstream and downstream to EDC. We excluded from the EDC group the S100 proteins, because they are not unique to differentiated keratinocytes; therefore, they can be

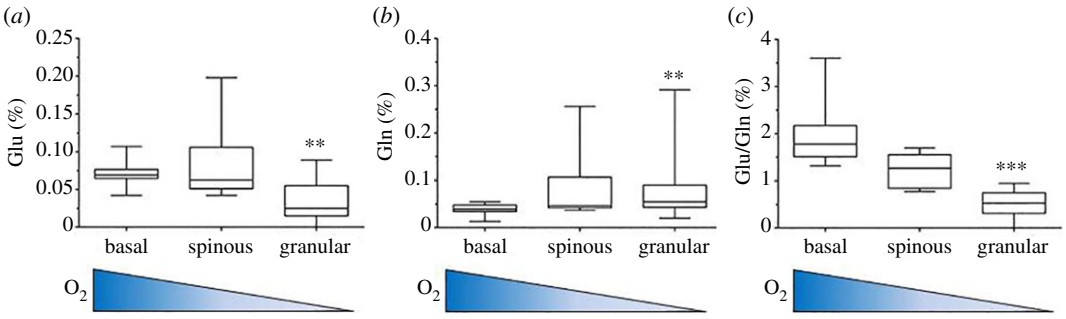

**Figure 2.** The content of glutamate, glutamine and the ratio Glu/Gln of protein markers of three epidermal layers. Box plots of values for proteins of each layer. Bottoms and tops of the boxes are the 25th and 75th percentiles, respectively; the lines across the boxes are the median values, the ends of the whiskers represent minimum and maximum values. The oxygen gradient throughout the layers is shown at the bottom of the graphs. **$p < 0.01$ compared to the basal layer. ***$p < 0.001$ compared to the basal layer.

transcribed and expressed in several different conditions [34]. The complete list of proteins analysed is shown in electronic supplementary material, table S1. The results of our evaluation are shown in figure 3. Compared with the other two groups of proteins, the EDC proteins have significantly less glutamate (figure 3a) and more glutamine (figure 3b); therefore, the ratio Glu/Gln is significantly the lowest in the EDC group (figure 3c), whereas the values of the other groups are spread out; the low ratio is in agreement with the values found in the granular layer, where these proteins are expressed.

We found another gene clustering that is transcribed only in granular keratinocytes. In fact, several lipase enzymes are codified by genes located in chromosome 10q23.31. This specialized human genomic locus includes six lipase genes and five other genes of apparently unrelated function. The human lipase genes appear to be exclusively expressed in the epidermis and their expression is highly specific for granular keratinocytes, which are exposed to the lowest oxygen tension of epidermis (7 mm Hg) [35]. The analysis of the human 10q23.31 locus originated a list of proteins that is shown in electronic supplementary material, table S2. The ratio Glu/Gln of the proteins codified by the lipase locus (LIP) was significantly lower than the ratio calculated for the proteins codified by the nearby genes, in agreement with the analysis of the EDC locus (figure 3e,f).

These data not only reinforce the results of the analysis of epidermal layers, but also they lead us to wonder whether the similar content in single AAs could be one of the reasons for gene clustering. In fact, this finding fits the principle that transcription and translation are coordinately driven by specific signals, among them the AA availability. In order to avoid a futile cycle of mRNA synthesis and demolition, transcription and translation must be matched. In bacteria, the RNA polymerase is closely followed by a ribosome that translates the newly synthesized transcript, thus the overall elongation rate of transcription is tightly controlled by the rate of translation [36]; such a cooperative mechanism ensures that the transcriptional activity is always adjusted to translational needs under various local conditions, such as AA availability. In eukaryotes, transcription and translation are spatially segregated, thus other mechanisms must control the matched transcription/translation. One pursued strategy is represented by the regulation of distinct gene sets through AA-sensitive enhancer regions AAREs (amino acid response elements). A multiproteic complex is bound to the AARE sequences and is involved in either inducing or repressing transcription of target genes in response to AA starvation [37–40]; in this way, many genes sparse in the genome can be read and produce proteins only when the proper levels of single AAs are reached. Although this mechanism of control is exploited by some AAs, glutamine does not seem to be involved in AARE regulation. In fact, glutamine is described as an important regulator of gene expression through the activation of transcription factors, without any evidence for a 'glutamine-responsive element' [41]. For example, the work by Bellon et al. [42] concluded that glutamine exerts its stimulating effect on collagen synthesis indirectly at the transcriptional level.

Another strategy to harmonize transcription and translation could be the proximal location of genes that must respond to the same environmental cues. In the mammalian genome, functionally related genes often are clustered in loci. Genes involved in execution of keratinocyte-specific gene expression programmes are clustered in at least four regions, including the epidermal differentiation complex (EDC), keratin clusters (type I and type II loci) and lipase cluster. The molecular mechanisms involved in coordinated gene regulation at these loci remain largely unknown. While keratins are employed in

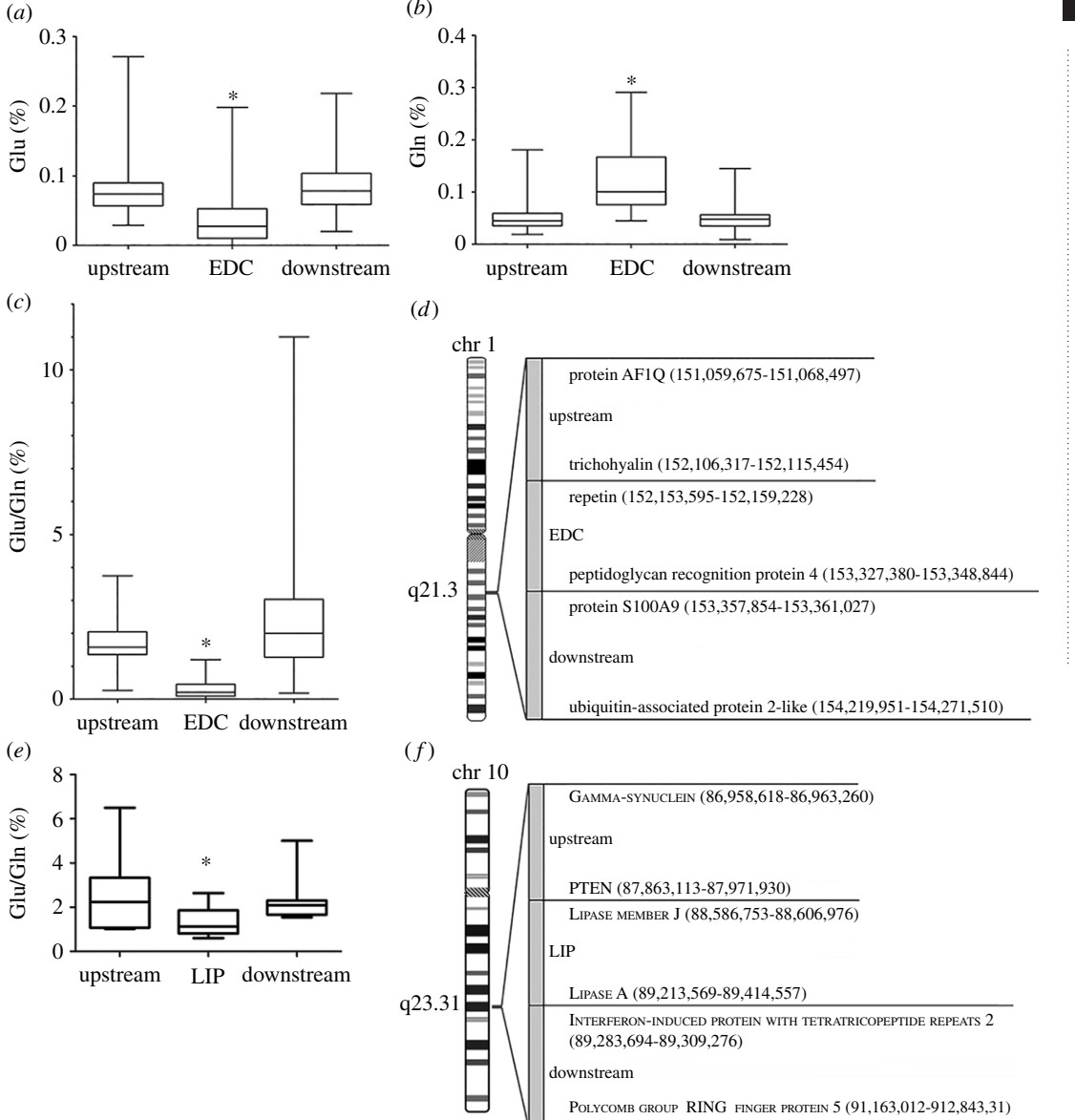

**Figure 3.** (*a*−*c*) The content of glutamate, glutamine and the ratio Glu/Gln of proteins coded by EDC and nearby regions of chromosome 1. (*e*) The ratio Glu/Gln of proteins coded by lipase cluster (LIP) and nearby regions of chromosome 10. Box plots of values for proteins of each region. Bottoms and tops of the boxes are the 25th and 75th percentiles, respectively; the lines across the boxes are the median values, the ends of the whiskers represent minimum and maximum values. *$p < 0.05$ compared to upstream and downstream regions. (*d*) Schematic of the chromosome 1q21.3 and (*f*) the chromosome 10q23.31. Border genes of the loci, the downstream and the upstream regions are indicated, and numbers (bp) show the start and end position of each gene on the chromosome [Ref: https://www.ncbi.nlm.nih.gov/genome/gdv/].

a large variety of tissues, the proteins coded by EDC and some lipases are specifically produced in skin; therefore, they were analysed in our study to demonstrate the validity of the ratio Glu/Gln as predictor of the oxygenated cellular context. Based on our analysis, it is reasonable to hypothesize that the AA availability could select and drive the simultaneous translation of many proteins, hence the evolutionary pressure to bring the genes together. Other loci could be sensitive to other AA availability, and further studies are warranted.

## 3.5. The liver model

The second model of tissue showing a gradient of oxygenation analysed in our study was the liver. In order to accomplish the complex functions necessary to maintain the metabolic homeostasis of the

**Table 2.** Selection of liver proteins divided into periportal and perivenous regions. Protein identity (ID) is indicated as found in UniProtKB/Swiss-Prot and the chromosomal location of the relative gene (Chr) is shown.

| layer | protein | ID UniProt | Chr | Glu/Gln (%) | Glu (%) | Gln (%) |
|---|---|---|---|---|---|---|
| periportal | collagen 4A1 | CO4A1_HUMAN | 13q34 | 0.959 | 0.042 | 0.044 |
| periportal | collagen 4A2 | CO4A2_HUMAN | 13q34 | 0.968 | 0.035 | 0.036 |
| periportal | collagen 4A3 | CO4A3_HUMAN | 2q36.3 | 1.396 | 0.04 | 0.029 |
| periportal | collagen 4A4 | CO4A4_HUMAN | 2q36.3 | 1.208 | 0.034 | 0.028 |
| periportal | collagen 4A5 | CO4A5_HUMAN | Xq22.3 | 0.836 | 0.036 | 0.043 |
| periportal | collagen 4A6 | CO4A6_HUMAN | Xq22.3 | 0.877 | 0.034 | 0.038 |
| periportal | collagen 5A1 | CO5A1_HUMAN | 9q34.3 | 1.622 | 0.065 | 0.04 |
| periportal | collagen 5A2 | CO5A2_HUMAN | 2q32.2 | 1.267 | 0.051 | 0.04 |
| periportal | collagen 5A3 | CO5A3_HUMAN | 19p13.2 | 1.295 | 0.058 | 0.045 |
| periportal | laminin lam A3 | LAMA3_HUMAN | 18q11.2 | 1.209 | 0.057 | 0.047 |
| periportal | laminin lam A4 | LAMA4_HUMAN | 6q21 | 1.57 | 0.068 | 0.043 |
| periportal | laminin lam B3 | LAPM5_HUMAN | 1q32.2 | 1.714 | 0.046 | 0.027 |
| periportal | laminin lam C3 | LAMC3_HUMAN | 9q34.12 | 1.029 | 0.068 | 0.066 |
| periportal | enzyme SULT1A1 | ST1A1_HUMAN | 16p11.2 | 1.692 | 0.075 | 0.044 |
| perivenous | collagen1A1 | CO1A1_HUMAN | 17q21.33 | 1.531 | 0.051 | 0.033 |
| perivenous | collagen1A2 | CO1A2_HUMAN | 7q21.3 | 2 | 0.048 | 0.024 |
| perivenous | collagen3A1 | CO3A1_HUMAN | 2q32.2 | 1.762 | 0.05 | 0.029 |
| perivenous | fibronectin | FINC_HUMAN | 2q35 | 1.085 | 0.059 | 0.054 |
| perivenous | glutamine synthetase | GLNA_HUMAN | 1q25.3 | 2.25 | 0.072 | 0.032 |
| perivenous | enzyme UGT1A1 | UD11_HUMAN | 2q37.1 | 1.263 | 0.045 | 0.036 |
| perivenous | enzyme UGT1A3 | UD13_HUMAN | 2q37.1 | 2 | 0.049 | 0.024 |
| perivenous | enzyme UGT1A4 | UD14_HUMAN | 2q37.1 | 2 | 0.049 | 0.024 |
| perivenous | enzyme UGT1A5 | UD15_HUMAN | 2q37.1 | 1.786 | 0.047 | 0.026 |
| perivenous | enzyme UGT1A6 | UD16_HUMAN | 2q37.1 | 2 | 0.053 | 0.026 |
| perivenous | enzyme UGT1A7 | UD17_HUMAN | 2q37.1 | 2.8 | 0.053 | 0.019 |
| perivenous | enzyme UGT1A8 | UD18_HUMAN | 2q37.1 | 2.333 | 0.053 | 0.023 |
| perivenous | enzyme UGT1A9 | UD19_HUMAN | 2q37.1 | 2.9 | 0.055 | 0.019 |
| perivenous | enzyme UGT1A10 | UD110_HUMAN | 2q37.1 | 2 | 0.053 | 0.026 |

whole organism, the liver is characterized by a structural and functional heterogeneity known as metabolic zonation [29]. The nutrients are supplied by the portal vein and the hepatic artery, and the blood leaves the organ via the central vein. The region around the portal vein is called periportal, and the area around the central vein is known as perivenous zone. Several concentration gradients of nutrients, substrates and hormones are established between zones, and depending on their position, the hepatocytes lined up along the porto-central axis are specialized in terms of metabolism, protein expression and function. The oxygen tension in the periportal zone is about 65 mm Hg and falls to about 35 mm Hg in the perivenous zone. Apparently, this could be another good model to study the relationship between Glu/Gln content of proteins and oxygenation of the cells; however, in the liver, the oxygen gradient is associated with a gradient of AA uptake, metabolism and secretion. In fact, part of the AA supply is catabolized to fuel the urea cycle; therefore, the specialized AA metabolism of the different zones influences the availability of AAs to synthesize proteins. Some catabolic enzymes are found preferentially expressed in the periportal zone, where they might support gluconeogenesis as well as urea synthesis [16], among them the enzyme glutaminase. In contrast, GS

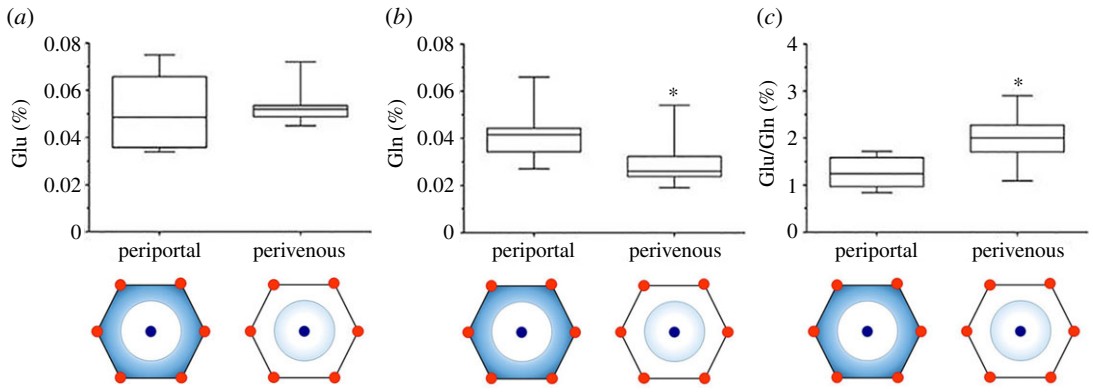

**Figure 4.** The content of glutamate, glutamine and the ratio Glu/Gln of periportal and perivenous proteins of the liver. Box plots of values for proteins of each zone. Bottoms and tops of the boxes are the 25th and 75th percentiles, respectively; the lines across the boxes are the median values, the ends of the whiskers represent minimum and maximum values. $*p < 0.05$ compared to periportal proteins. The most oxygenated area of the lobular structure (periportal) and the less oxygenated zone (perivenous) are schematically depicted at the bottom of the graphs.

is located in the perivenous area of the liver lobule, which has a secretory function. The intercellular glutamine cycling facilitates urea production: portal glutamine is catabolized by the enzyme glutaminase to increase periportal urea synthesis, and glutamine is reconstituted by perivenous synthesis through GS to replenish blood levels of this AA [15]. Based on this biochemical evaluation of hepatic tissue, when analysing the protein composition, we predicted a different result to what was found in the epidermis, since in the liver, the availability of AAs to build proteins is affected not only by the oxygen gradient but also and foremost, it is influenced by the homeostatic function of the organ that distributes the nutrients to the whole body.

We selected some proteins described in the literature as preferentially expressed in the periportal or the perivenous zone of the liver and we carried out the analysis of the levels of Glu, Gln and their ratio. The list of the proteins evaluated is reported in table 2. Considering that the periportal zone is more oxygenated than the perivenous zone, we found that the ratio Glu/Gln was not directly proportional to oxygenation, since it was higher in the perivenous zone, as shown in figure 4. This was due to the decrease in Gln content in proteins of perivenous zone, whereas the availability of glutamate was constant in both zones. Such a decrease is explained by the fact that glutamine is mostly taken and consumed by periportal epatocytes to feed the urea cycle; therefore, the perivenous epatocytes do not receive glutamine; moreover, they synthesize glutamine but only to replenish the serum levels of the AA and not for local utilization. It is of note that GS is expressed in the perivenous hypoxic zone; hence, we found confirmation that also in the liver, the hypoxic context upregulates GS; however, the glutamine is produced and secreted; therefore, the availability of glutamine to build intracellular proteins is low.

## 3.6. The ratio Glu/Gln in protein expression profile as adaptation to oxygen levels

In the epidermis model, we have analysed the Glu/Gln ratio of different proteins in the same tissue exposed to an oxygen gradient and we have demonstrated that the Glu/Gln ratio correlates with the oxygenation. We can transpose the concept to the analysis of the same protein produced as isoform in distinct tissues differently oxygenated or to the comparison of proteins that alternate their expression in the same tissue with an oxygen-related oscillation. As examples of the first application, we investigated two proteins that are expressed as isoforms in different tissues. Lactate dehydrogenase (LDH) is a tetramer made up of two different subunits A and B. The ratio between A and B depends on local conditions. Usually, LDH A is associated with lower $pO_2$ and low $NAD^+$ and catalyses the reaction from pyruvate to lactate, for example, in the muscle mostly during anaerobic work. LDH B isoform operates in well-oxygenated tissues, for example, heart, in the opposite reaction. Low oxygen availability upregulates LDH A [43] and downregulates LDH B expression [44]. Steroid 5-α-reductase (S5A) converts testosterone into 5-α-dihydrotestosterone and progesterone or corticosterone into their corresponding 5-α-3-oxosteroids. S5A1 is the isoform expressed in the brain [45] (average oxygen tension at 33.8 mm Hg [46]), whereas the isoform 2 (S5A2) is abundant in prostate [47] (hypoxic tissue with the average oxygen pressure at 4.5 mm Hg [48]).

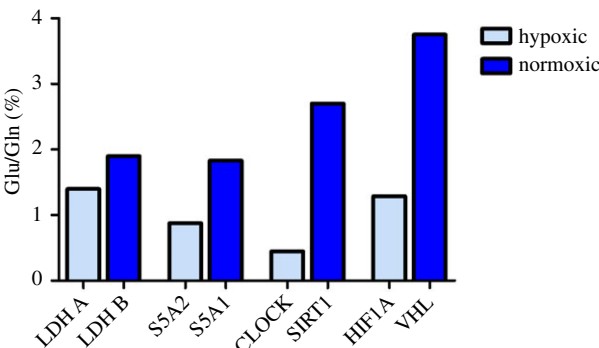

**Figure 5.** The ratio Glu/Gln of proteins selectively expressed in hypoxic or normoxic environment: LDH isoforms A and B, 5-α-reductase (S5A) isoforms 1 and 2, Circadian Locomotor Output Cycles Kaput (CLOCK), sirtuin 1 (SIRT1), HIF1A, VHL tumour suppressor.

Among the second type of proteins with an oscillatory reciprocal control, we considered the CLOCK/SIRT1 and HIF1α/von Hippel–Lindau (VHL) protein pairs. The levels of oxygen vary during the day in many tissues and change the expression pattern of many proteins [49]. CLOCK has an intrinsic acetyltransferase activity, which enables circadian chromatin remodelling, and sirtuin 1 (SIRT1) is a deacetylase involved in transcriptional silencing that antagonizes CLOCK activity [50]. Because its activity is NAD+-dependent, SIRT1 must operate in high oxygen conditions, when the ratio NAD+/NADH is high; indeed, SIRT1 is downregulated in hypoxia [51]. Finally, another pair of proteins plays antagonistic roles: hypoxia-inducible factor 1-α (HIF1A) is a well-known mediator of hypoxia, and VHL tumour suppressor is an ubiquitin ligase capable of targeting the HIFα subunits at normoxia for destruction by the proteasome.

All these proteins were analysed for their Glu/Gln ratio and we found that in each pair, the ratio was proportional to oxygen levels (figure 5). These data are in agreement with the results obtained from the analysis of the epidermal model and reinforce the relevance of our study, which demonstrates that it is possible to predict whether a protein is produced in a normoxic or hypoxic environment based on its Glu/Gln content.

## 4. Conclusion

In this work, we tested the hypothesis that the AA composition of proteins reflects the availability of AA imposed by the cellular context: in particular, we considered that the oxygenated environment would advantage the biosynthesis of proteins enriched in glutamate, while the hypoxic condition would increase the availability of glutamine and thus favour the translation of glutamine-rich proteins.

In this study, we worked on tissues that display a gradient of oxygen and we demonstrated that when the tissue has a local utilization of AA, the ratio Glu/Gln is proportional to oxygenation.

The first tissue model analysed supported our working hypothesis. We demonstrated that the ratio Glu/Gln of the epidermal proteins was directly proportional to the oxygenation of the layer expressing the protein, evidently due to a conversion of glutamate to glutamine driven by the hypoxic context. The analysis of the EDC and lipase locus not only further validated these conclusions, but interestingly, it also suggested that gene clustering may represent an adaptation for responding to AA availability.

The principle demonstrated in the epidermal model, namely that the ratio Glu/Gln is directly proportional to oxygenation, holds true if the AA uptake does not change between the considered areas and the AA are incorporated only in endogenous proteins. We hypothesized that if other metabolic gradients relying on glutamine utilization were present, the concentration of glutamate and glutamine would be the result of a variable uptake, biosynthesis and loss of the AA; therefore, the levels of the two AAs would not be predictive of a local cellular context. This was verified in the liver model, where the ratio Glu/Gln was found inversely proportional to oxygenation due to the loss of glutamine in the most hypoxic area of the liver. Based on the comparison of the two models, we reckon that the ratio Glu/Gln and oxygenation are directly correlated only when the observed cells are self-centred systems.

Altogether, in this work, we found that the epidermis represents a simple model of metabolic layering, where the local utilization of AAs allows us to demonstrate that the ratio Glu/Gln found in

proteins is directly proportional to oxygen levels, but in other more complex models, all the variables must be considered.

Based on our analysis and considerations, the evaluation of Glu/Gln ratio in proteins could be exploited in the search for protein markers of the hypoxic microenvironment, which is common in several pathological conditions.

Data accessibility. Our data are included in the electronic supplementary material.
Authors' contributions. F.S. and G.P. constructed the biochemical theory and analysed the biochemistry of the proposed models. D.M. contributed to experimental design and data analysis. A.V. and C.R. supported the numerical computation of the biochemical model. All authors discussed the results at the all stages and contributed to the article drafting and revision. F.S. wrote the final version of the paper. All authors gave final approval for publication.
Competing interests. The authors declare that they have no conflict of interest.
Funding. This research did not receive any specific grant from funding agencies in the public, commercial or not-for-profit sectors.

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
