## [Reviewer comments · Royal Society Open Science]

Review History

RSOS-181891.R0 (Original submission)

Review form: Reviewer 1 (Ramon Sun)

Is the manuscript scientifically sound in its present form?

Yes

Are the interpretations and conclusions justified by the results?

Yes

Is the language acceptable?

Yes

Is it clear how to access all supporting data?

Not Applicable

Do you have any ethical concerns with this paper?

No

Have you any concerns about statistical analyses in this paper?

I do not feel qualified to assess the statistics

Recommendation?

Major revision is needed (please make suggestions in comments)

Comments to the Author(s)

"The analysis of glutamate and glutamine frequencies in human proteins as markers of tissue oxygenation"

Summary:

The authors of this study examined the ratio of glutamate to glutamine in proteins, and related these values to local oxygenation status in the environment where the protein is produced. The tissues selected in this study, epidermis and liver, were chosen because both are striated in terms of access to nutrients and oxygenation status, and thus could be evaluated by use of tissue/zone specific protein expression. Because glutamine synthesis is favored under conditions of hypoxia, the authors hypothesized that regions of tissue that have less access to oxygen would preferentially produce proteins containing more glutamine (i.e. the Glu:Gln ratio would be lower). To test this hypothesis, the authors used cluster and statistical analysis of Gln:Glu ratios in proteins expressed in each zone of the epidermis or liver. Their hypothesis was supported by the epidermis model in which metabolism may be more self-contained, but the more complicated metabolic requirements of the liver indicated that the hypoxia model may only serve as a good predictor of protein expression in such self-contained systems.

Overall, the authors generate the interesting hypothesis that the known effects of hypoxia on glutamine synthetase and glutaminase, which favor higher glutamine levels, ultimately result in the translation of proteins reflective of oxygenation status. While the link between hypoxia and glutamine metabolism has been previously demonstrated, the link between glutamine/glutamate availability and protein translation seems more tenuous. The authors suggest that this link could be 1) amino acid response elements (AAREs) that function as enhancers that sense AA levels, and 2) spatial organization of genes with similar AA compositions. Both of these are reasonable suggestions, but neither is explored beyond speculation. Moreover, the relationship that the authors are trying to establish is further complicated by tissue access to external AA sources, which is acknowledged in the liver model. Ultimately, this study establishes a correlation between oxygen levels and preferred protein translation in specific model systems, but provides no causal link.

Major points:

- 1) The link between AA availability and protein translation could be greatly strengthened by the identification of AAREs upstream of the analyzed proteins in both epidermis and liver models. If such enhancers are identified, it could help resolve the conflicting trend found in liver tissue.
- 2) The authors suggest that genes might be physically clustered based on AA composition, so that they can be translated together when a specific AA is abundant. Besides the EDC, are there other such clusterings that could be identified? This, again, would support the link between hypoxia and protein AA content.
- 3) As the authors demonstrate, hypoxia is only a valid predictor of preferential protein production when metabolism is relatively self-contained. Are there other such systems that could further support the epidermis model? I.e., where else is this model useful? The hypothesis could

be made more universal if Major Point 5 is addressed.

4) The authors, as they acknowledge, evaluate protein AA content in terms of percentages, which removes protein size from consideration. It would be interesting if the authors also performed an evaluation on absolute numbers of Glu/Gln within a protein and its expression level. For example, would a large protein with a much higher number of glutamines (even if the ratio Glu:Gln is high) be translated more under conditions of hypoxia than a smaller protein with a similar AA content? This would strengthen their hypothesis considerably and support the trend seen in their evaluation of Glu/Gln ratios.

5) Ultimately, the link between hypoxia and protein translation will need to be tested experimentally. The authors could conceivably test this hypothesis in a cell-culture system by varying levels of glutamine and evaluating transcription of target genes by RT-PCR. Until an experiment like this is done, the relationship at the heart of this manuscript will remain speculation.

6) The authors say on page 4, line 26-28, that: "The exposure to hypoxia increases both GS mRNA, protein levels and enzymatic activity..." Where is GS mRNA increased? It seems important to this study that mRNA levels of GS are increased in the two model systems presented.

Minor points:

1) If data is available to quantify oxygen levels in epidermal layers, it would be better than "low, medium, and high."

2) All references beyond reference 31 are missing from the reference list (but still cited in paper).

3) References are not cited consistently – some are cited in parenthesis, some as superscript, and some with author name.

4) The sentence on page 3, lines 16-26, is confusing and could be written much more clearly.

5) The sentence on page 11, line 22: "the overall elongation rate of transcription is tightly controlled by translation" seems as if the words transcription/translation were swapped. This could not be verified, however, as reference 32 was not provided.

Review form: Reviewer 2 (Roberto Scatena)

Is the manuscript scientifically sound in its present form?

Yes

Are the interpretations and conclusions justified by the results?

Yes

Is the language acceptable?

Yes

Is it clear how to access all supporting data?

Yes

Do you have any ethical concerns with this paper?

No

Have you any concerns about statistical analyses in this paper?

No

Recommendation?

Accept as is

Comments to the Author(s)

None

Decision letter (RSOS-181891.R0)

23-Jan-2019

Dear Dr Silvagno,

The editors assigned to your paper ("The analysis of glutamate and glutamine frequencies in human proteins as marker of tissue oxygenation") have now received comments from two independent reviewers. They expressed interest in your work but also raised some concerns. Hence, we would like you to revise your paper in accordance with the referee and Associate Editor suggestions which can be found below (not including confidential reports to the Editor). Please note this decision does not guarantee eventual acceptance.

Please submit a copy of your revised paper before 15-Feb-2019. Please note that the revision deadline will expire at 00.00am on this date. If we do not hear from you within this time then it will be assumed that the paper has been withdrawn. In exceptional circumstances, extensions may be possible if agreed with the Editorial Office in advance. We do not allow multiple rounds of revision so we urge you to make every effort to fully address all of the comments at this stage. If deemed necessary by the Editors, your manuscript will be sent back to one or more of the original reviewers for assessment. If the original reviewers are not available, we may invite new reviewers.

- Data accessibility

It is a condition of publication that all supporting data are made available either as supplementary information or preferably in a suitable permanent repository. The data

accessibility section should state where the article's supporting data can be accessed. This section should also include details, where possible of where to access other relevant research materials such as statistical tools, protocols, software etc can be accessed. If the data have been deposited in an external repository this section should list the database, accession number and link to the DOI for all data from the article that have been made publicly available. Data sets that have been deposited in an external repository and have a DOI should also be appropriately cited in the manuscript and included in the reference list.

If you wish to submit your supporting data or code to Dryad (<http://datadryad.org/>), or modify your current submission to dryad, please use the following link:
<http://datadryad.org/submit?journalID=RSOS&manu=RSOS-181891>

- **Competing interests**

- **Authors' contributions**

- **Acknowledgements**

- **Funding statement**

on behalf of Professor Erika Mancini (Associate Editor) and Professor Katrin Rittinger (Subject Editor)
openscience@royalsociety.org

Comments to Author:

Reviewers' Comments to Author:

Reviewer: 1

Comments to the Author(s)

“The analysis of glutamate and glutamine frequencies in human proteins as markers of tissue oxygenation”

Summary:

The authors of this study examined the ratio of glutamate to glutamine in proteins, and related these values to local oxygenation status in the environment where the protein is produced. The tissues selected in this study, epidermis and liver, were chosen because both are striated in terms of access to nutrients and oxygenation status, and thus could be evaluated by use of tissue/zone specific protein expression. Because glutamine synthesis is favored under conditions of hypoxia, the authors hypothesized that regions of tissue that have less access to oxygen would preferentially produce proteins containing more glutamine (i.e. the Glu:Gln ratio would be lower). To test this hypothesis, the authors used cluster and statistical analysis of Gln:Glu ratios in proteins expressed in each zone of the epidermis or liver. Their hypothesis was supported by the epidermis model in which metabolism may be more self-contained, but the more complicated metabolic requirements of the liver indicated that the hypoxia model may only serve as a good predictor of protein expression in such self-contained systems.

Overall, the authors generate the interesting hypothesis that the known effects of hypoxia on glutamine synthetase and glutaminase, which favor higher glutamine levels, ultimately result in the translation of proteins reflective of oxygenation status. While the link between hypoxia and glutamine metabolism has been previously demonstrated, the link between glutamine/glutamate availability and protein translation seems more tenuous. The authors suggest that this link could be 1) amino acid response elements (AAREs) that function as enhancers that sense AA levels, and 2) spatial organization of genes with similar AA compositions. Both of these are reasonable suggestions, but neither is explored beyond speculation. Moreover, the relationship that the authors are trying to establish is further complicated by tissue access to external AA sources, which is acknowledged in the liver model. Ultimately, this study establishes a correlation between oxygen levels and preferred protein translation in specific model systems, but provides no causal link.

Major points:

- 1) The link between AA availability and protein translation could be greatly strengthened by the identification of AAREs upstream of the analyzed proteins in both epidermis and liver models. If such enhancers are identified, it could help resolve the conflicting trend found in liver tissue.
- 2) The authors suggest that genes might be physically clustered based on AA composition, so that they can be translated together when a specific AA is abundant. Besides the EDC, are there other such clusterings that could be identified? This, again, would support the link between hypoxia and protein AA content.
- 3) As the authors demonstrate, hypoxia is only a valid predictor of preferential protein production when metabolism is relatively self-contained. Are there other such systems that could further support the epidermis model? I.e., where else is this model useful? The hypothesis could be made more universal if Major Point 5 is addressed.
- 4) The authors, as they acknowledge, evaluate protein AA content in terms of percentages, which removes protein size from consideration. It would be interesting if the authors also performed an

evaluation on absolute numbers of Glu/Gln within a protein and its expression level. For example, would a large protein with a much higher number of glutamines (even if the ratio Glu:Gln is high) be translated more under conditions of hypoxia than a smaller protein with a similar AA content? This would strengthen their hypothesis considerably and support the trend seen in their evaluation of Glu/Gln ratios.

5) Ultimately, the link between hypoxia and protein translation will need to be tested experimentally. The authors could conceivably test this hypothesis in a cell-culture system by varying levels of glutamine and evaluating transcription of target genes by RT-PCR. Until an experiment like this is done, the relationship at the heart of this manuscript will remain speculation.

6) The authors say on page 4, line 26-28, that: "The exposure to hypoxia increases both GS mRNA, protein levels and enzymatic activity..." Where is GS mRNA increased? It seems important to this study that mRNA levels of GS are increased in the two model systems presented.

Minor points:

- 1) If data is available to quantify oxygen levels in epidermal layers, it would be better than "low, medium, and high."
- 2) All references beyond reference 31 are missing from the reference list (but still cited in paper).
- 3) References are not cited consistently – some are cited in parenthesis, some as superscript, and some with author name.
- 4) The sentence on page 3, lines 16-26, is confusing and could be written much more clearly.
- 5) The sentence on page 11, line 22: "the overall elongation rate of transcription is tightly controlled by translation" seems as if the words transcription/translation were swapped. This could not be verified, however, as reference 32 was not provided.

Reviewer: 2

Comments to the Author(s)

This interesting article deal with the analysis of amino acid content as indicator of a specific cellular context. Specifically, the article stresses the role of oxygen gradient on the control of protein expression. It represent a new approach to evaluate general interest in biological chemistry and estimation of glutamate/glutamine ratio can give useful information on the level of tissue oxygenation. This could be a potential marker of hypoxia, a common condition in the pathophysiology of various diseases.

The manuscript appears well written and protocol well developed.

Author's Response to Decision Letter for (RSOS-181891.R0)

See Appendix A.

Decision letter (RSOS-181891.R1)

07-Mar-2019

Dear Dr Silvagno,

I am pleased to inform you that your manuscript entitled "The analysis of glutamate and glutamine frequencies in human proteins as marker of tissue oxygenation" is now accepted for publication in Royal Society Open Science.

on behalf of Professor Erika Mancini (Associate Editor) and Katrin Rittinger (Subject Editor)
openscience@royalsociety.org

Reviewer comments to Author:

Appendix A

Reviewers' Comments to Author:

Reviewer: 1

Comments to the Author(s)

“The analysis of glutamate and glutamine frequencies in human proteins as markers of tissue oxygenation”

Summary:

The authors of this study examined the ratio of glutamate to glutamine in proteins, and related these values to local oxygenation status in the environment where the protein is produced. The tissues selected in this study, epidermis and liver, were chosen because both are striated in terms of access to nutrients and oxygenation status, and thus could be evaluated by use of tissue/zone specific protein expression. Because glutamine synthesis is favored under conditions of hypoxia, the authors hypothesized that regions of tissue that have less access to oxygen would preferentially produce proteins containing more glutamine (i.e. the Glu:Gln ratio would be lower). To test this hypothesis, the authors used cluster and statistical analysis of Gln:Glu ratios in proteins expressed in each zone of the epidermis or liver. Their hypothesis was supported by the epidermis model in which metabolism may be more self-contained, but the more complicated metabolic requirements of the liver indicated that the hypoxia model may only serve as a good predictor of protein expression in such self-contained systems.

Overall, the authors generate the interesting hypothesis that the known effects of hypoxia on glutamine synthetase and glutaminase, which favor higher glutamine levels, ultimately result in the translation of proteins reflective of oxygenation status. While the link between hypoxia and glutamine metabolism has been previously demonstrated, the link between glutamine/glutamate availability and protein translation seems more tenuous. The authors suggest that this link could be 1) amino acid response elements (AAREs) that function as enhancers that sense AA levels, and 2) spatial organization of genes with similar AA compositions. Both of these are reasonable suggestions, but neither is explored beyond speculation.

Author response: The suggested mechanisms are discussed as two explanatory models for gene clustering. In fact we analysed the EDC complex, which is responsible for the coordinated synthesis of many proteins enriched in glutamine. Our belief is that the amino acid availability selects and drives the simultaneous translation of many proteins, hence the evolutionary pressure to bring the genes together, as we discuss at page 12. Because this is a novel hypothesis, we considered possible and discussed also the possibility supported by previous studies, which is the presence of AAREs sensing the amino acid levels and enhancing the coordinated protein synthesis. Stimulated by the comments of the

Reviewer, we went further into the subject and we found out that the glutamine does not seem to be involved in AARE regulation, differently from other amino acids. We added this observation in the text.

Moreover, the relationship that the authors are trying to establish is further complicated by tissue access to external AA sources, which is acknowledged in the liver model. Ultimately, this study establishes a correlation between oxygen levels and preferred protein translation in specific model systems, but provides no causal link.

Author response: In our opinion the causal link is the control of translation exerted by glutamine availability. Whereas other amino acid can directly control transcription through AAREs, no one ever found evidences that glutamine works this way. Besides the control of transcription, other mechanisms can influence the production of proteins, such as mRNA stability and translation. Glutamine could work controlling the steps that follow transcription, which should be reasonably linked. For example when the translation of a gln-rich protein slows down because of scarce availability of glutamine, the stability of messenger is affected and the synthesis of the protein is reduced, and vice versa.

Major points:

1) *The link between AA availability and protein translation could be greatly strengthened by the identification of AAREs upstream of the analyzed proteins in both epidermis and liver models. If such enhancers are identified, it could help resolve the conflicting trend found in liver tissue.*

Author response: We believe that the conflicting trend is due to intracellular availability of gln, as we discussed in the manuscript, page 16. We propose that the difference between skin and liver is not generated by a conflicting control of transcription but rather by a different utilization of glutamine, because the amino acid is incorporated into proteins by skin and released in circulation by liver. Moreover, glutamine does not seem to be involved in AARE regulation, differently from other amino acids. Glutamine is described as an important regulator of gene expression without any evidence for a 'glutamine- responsive element'. This point has been added in text at page 12.

Based on these considerations, the research of AAREs would give results not linked to the ratio glu/gln of the analysed proteins.

2) *The authors suggest that genes might be physically clustered based on AA composition, so that they can be translated together when a specific AA is abundant. Besides the EDC, are there other such clusterings that could be identified? This, again, would support the link between hypoxia and protein AA content.*

Author response: After further evaluation of published data, we found another gene clustering that is transcribed only in granular keratinocytes: the family of lipase enzymes, which are codified by genes located in chromosome 10 q23.31. This specialized human genomic locus includes six lipase genes and five other genes of

apparently unrelated function. The human *lipase* genes appear to be exclusively expressed in the epidermis and their expression is highly specific for granular keratinocytes that are exposed to the lowest oxygen tension of epidermis (7 mmHg). The analysis of the human 10q23.31 locus originated a list of proteins that is shown in supplementary Table S2. The ratio glu/gln of the lipase locus (LIP) was significantly lower than the ratio calculated for the proteins codified by the nearby genes, in agreement with the analysis of the EDC locus. The new data have been added in Fig.3 and are discussed at page 11.

3) *As the authors demonstrate, hypoxia is only a valid predictor of preferential protein production when metabolism is relatively self-contained. Are there other such systems that could further support the epidermis model? I.e., where else is this model useful? The hypothesis could be made more universal if Major Point 5 is addressed.*

Author response: In this work we have analysed the glu/gln ratio of different proteins in the same tissue exposed to an oxygen gradient. This model is unique in its simplicity and self-containment. However, we can transpose the concept when we compare the glu/gln ratio of the same protein produced as isoform in distinct tissues differently oxygenated or when we compare proteins that alternate their expression in the same tissue with an oxygen-related oscillation. A new analysis has been added as a new paragraph at page 14, with an additional figure 5.

4) *The authors, as they acknowledge, evaluate protein AA content in terms of percentages, which removes protein size from consideration. It would be interesting if the authors also performed an evaluation on absolute numbers of Glu/Gln within a protein and its expression level. For example, would a large protein with a much higher number of glutamines (even if the ratio Glu:Gln is high) be translated more under conditions of hypoxia than a smaller protein with a similar AA content? This would strengthen their hypothesis considerably and support the trend seen in their evaluation of Glu/Gln ratios.*

Author response: The proposed evaluation would be interesting, actually when we planned our analysis we took into consideration this approach. However, there are a number of different obstacles that can interfere with this kind of analysis. First of all, experimental data comparing the expression of the selected proteins in the same tissue in the same experimental conditions are not available, therefore we do not know whether the size and the expression levels are related. Second, many steps following transcription could influence the amount of protein synthesized and could introduce unpredictable variables, for example the stability of the mRNA or the amino acid sequence in the protein. We demonstrate that the ratio glu/gln is important, but other factors could interfere with the translation process and the relationship between absolute numbers of Glu/Gln and protein expression could be not necessarily linear. Third, it is of note that the ratio Glu/Gln does not change between absolute numbers and percentage, and in our analysis we highlight the importance of the ratio in relation to oxygen levels. For these reasons, we decided to work with percentages and removed protein size from consideration. We added a comment at page 9.

5) *Ultimately, the link between hypoxia and protein translation will need to be tested experimentally. The authors could conceivably test this hypothesis in a cell-*

culture system by varying levels of glutamine and evaluating transcription of target genes by RT-PCR. Until an experiment like this is done, the relationship at the heart of this manuscript will remain speculation.

Author response: The aim of our work was to analyse a self-contained model to prove the correlation between glu/gln ratio and tissue oxygenation. Once this is proved beyond speculation, some hypothesis can be put forward about the mechanisms, and it is true that this study, not conceived as an experimental work, cannot reach definitive conclusions. We assume that glutamine does not modulate directly the transcription step, because it has been reported that it does not work through AAREs and because we could not find in literature evidences of mRNAs levels modulated directly by glutamine. For example the work by Bellon and coll. concluded that glutamine exerts its stimulating effect on collagen synthesis indirectly at the transcriptional level, with a mechanism requiring protein synthesis. If glutamine controls the activity of many transcription factors, the variation of its levels in vitro could influence many pathways and the results could be unclear. Moreover, we expect that the deprivation of gln would induce glutamine synthase activity, as shown by the studies in vitro on keratinocytes carried out by Danielyan (reference 8), and this would override the modulation of transcription. For these reasons we feel that the suggested experiments would not give definitive answers about the mechanism that links glutamine availability and glutamine incorporation into proteins. We added a comment at page 12.

6) *The authors say on page 4, line 26-28, that: “The exposure to hypoxia increases both GS mRNA, protein levels and enzymatic activity...” Where is GS mRNA increased? It seems important to this study that mRNA levels of GS are increased in the two model systems presented.*

Author response: The references cited in the text refer to the increase of GS activity induced by hypoxia in vivo, in brain (ref 8), muscle and liver (ref 9). One study in vitro (10) carried out on PC12 cells demonstrated the induction of mRNA, protein and activity.

We have modified the text to make the information clear.

The study of Danielyan and coll. (reference 8) showed the high expression of the protein in the epidermal granular layer by immunohistochemistry and the increased activity of GS in vitro when keratinocytes were deprived of glutamine or the same cells were treated with ammonium chloride or dexamethasone. It is therefore clear that during the passage from basal layer to outer layers GS is induced by hypoxia, surely also at the mRNA level although not directly measured. Interestingly, also in the liver model the same induction occurs in the hypoxic zone, as in table 2 GS is listed among the proteins specifically expressed in the perivenous hypoxic zone. This is highlighted also in the text at page 14. Therefore we can confirm that mRNA levels of GS must be increased in the two model systems presented.

Minor points:

1) *If data is available to quantify oxygen levels in epidermal layers, it would be better than “low, medium, and high.”*

Author response: We described the oxygen tension of different epidermal layers at page 8.

2) *All references beyond reference 31 are missing from the reference list (but still cited in paper).*

3) *References are not cited consistently—some are cited in parenthesis, some as superscript, and some with author name.*

Author response: We corrected the references, sorry for the previous mistakes.

4) *The sentence on page 3, lines 16-26, is confusing and could be written much more clearly.*

Author response: The sentence has been modified

5) *The sentence on page 11, line 22: “the overall elongation rate of transcription is tightly controlled by translation” seems as if the words transcription/translation were swapped. This could not be verified, however, as reference 32 was not provided.*

Author response: The sentence is correct and the reference has been revised.

Reviewer: 2

Comments to the Author(s)

This interesting article deal with the analysis of amino acid content as indicator of a specific cellular context. Specifically, the article stresses the role of oxygen gradient on the control of protein expression. It represent a new approach to evaluate general interest in biological chemistry and estimation of glutamate/glutamine ratio can give useful information on the level of tissue oxygenation. This could be a potential marker of hypoxia, a common condition in the pathophysiology of various diseases.

The manuscript appears well written and protocol well developed.